# Hepatitis C Virus Infection in Persons Who Inject Drugs in the Middle East and North Africa: Intervention Strategies

**DOI:** 10.3390/v13071363

**Published:** 2021-07-14

**Authors:** Jag H. Khalsa, Poonam Mathur

**Affiliations:** 1Medical Consequences of Drug Abuse and Infections Branch, Division of Therapeutics and Medical Consequences, National Institute on Drug Abuse, National Institutes of Health, Bethesda, MD 20892, USA; 2Institute of Human Virology, University of Maryland School of Medicine, Baltimore, MD 21201, USA; pmathur@ihv.umaryland.edu

**Keywords:** hepatitis C virus infection, substance use disorder, intervention strategies, Middle East, North Africa (MENA)

## Abstract

There is a high incidence and prevalence of hepatitis C viral infection in persons with or without substance use disorders (SUDs) in the Middle East and North Africa (MENA) region, but only a small number receive comprehensive care. Highly effective direct-acting antiviral (DAA) medications are available at substantially lower costs; however, complete elimination of the hepatitis C virus (HCV) can only be achieved if integrated care strategies target those at highest risk for HCV infection and transmission and improve access to care. Due to the high prevalence of SUD in the MENA region, strategies to eliminate HCV must focus on integrated healthcare across multiple subspecialties, including addiction medicine, psychiatry, infectious diseases, hepatology, and social work. In this invited manuscript, we review the epidemiology of HCV in the MENA region and highlight intervention strategies to attain the WHO’s goal of HCV eradication by 2030.

## 1. Introduction

Hepatitis C virus (HCV) is a blood-borne pathogen caused by a hepatotrophic, single-stranded ribonucleic acid (RNA) virus. The acute phase of HCV infection may last about six months and cause no symptoms; an estimated 15–25% of these people spontaneously clear the infection and recover. However, the remaining 75–85% develop chronic HCV infection. Even in chronic HCV, people may be asymptomatic for 25–30 years; however, 20% of these individuals eventually develop liver cirrhosis, hepatic decompensation, and/or hepatocellular carcinoma (HCC), which are associated with high morbidity and mortality [1,2,3]. In 2017, the World Health Organization (WHO) reported that an estimated 71 million people were infected with HCV globally [3], and, in 2016, approximately 399,000 people died from HCV, mostly due to cirrhosis and hepatocellular carcinoma. A major hurdle in determining global epidemiology is the discrepancy between various serology assays and the lack of confirmation of chronicity due to the requirement of expensive HCV RNA testing. However, recent validation studies conducted by the POLARIS Observatory determined a 1% seroprevalence of HCV prevalence globally [4]. They also defined HCV prevalence in populations at risk: the pooled mean rate ranged between 57.4% (95% CI: 49.4–65.2%) in people who inject drugs and 75.5% (95% CI: 61.0–87.6%) in populations with liver-related conditions [5].

Symptoms of untreated HCV are often extrahepatic, manifesting most commonly as arthralgia, paresthesia, myalgia, pruritus, sicca syndrome, and sensory neuropathy. However, untreated HCV can lead to progressive liver damage and fibrosis, causing advanced or decompensated liver disease. Chronic liver disease causes synthetic liver dysfunction and portal hypertension, which results in mental status changes (hepatic encephalopathy), volume overload, and varices. HCV is a leading cause of HCC in western countries and increases the risk of cancer 15–20-fold [6]. Successful eradication of HCV reduces the risk of HCC by 71–75% [7].

The treatment of HCV has evolved over the last 10 years, with the advent of oral direct-acting antivirals (DAAs) [8]. Prior to 2011, the mainstay of HCV treatment was pegylated interferon and ribavirin. These two medications had limited efficacy and numerous side effects; a systematic review and meta-analysis estimated that only 25% of patients with chronic HCV infection were treated with these therapies [9]. The cure rate associated with DAAs is approximately 98% [10], which is much higher than the 50% cure rate associated with interferon-based therapy [11]. Now that the potency of HCV treatment has improved, the focus has shifted to expanding access to treatment and decreasing HCV-associated morbidity and mortality. In 2016, the World Health Organization (WHO) adopted the *Global Health Sector Strategy on Viral Hepatitis, 2016–2021*. The strategy envisions eliminating hepatitis (B and C) as a global public health problem and commits to a WHO-led effort to eliminate hepatitis B and C by 2030 [3].

The WHO recognizes that mortality from HBV and HCV mainly persists due to poor access to treatment. Therefore, it is pertinent to focus on how to target marginalized populations who are at high risk for HCV acquisition and transmission. In particular, campaigns have focused on targeting people who inject drugs (PWIDs). Different regions of the world have identified that increasing HCV treatment access for PWIDs is crucial to achieving the WHO’s goal of HCV elimination in the next 9 years. In the United States, 80% of HCV transmission occurs among PWIDs [12]. In Europe, HCV prevalence in PWIDs is between 39.9% and 28.6%, whereas HCV prevalence in the general population is 1.5% [13]. In India, the HCV prevalence rate is 44.7% among PWIDs, almost double that of any other high-risk group, whereas the generalized community prevalence is 0.85% [14]. Among other Southeast Asian countries, it is recognized that injection drug use remains an important route of HCV transmission, with HCV infecting more PWIDs than HIV or hepatitis B [15].

Difficulties arise in identifying where the care cascade is failing in low- and middle-income countries (LMICs). Though data on these countries exist, systematic analyses and subsequent policies to guide strategic intervention are limited by large, national-level studies [14,15]. It is estimated that over 80% of the world’s HCV infections are in LMICs [16]. The Middle East and North Africa region, for example, is an area that numerous review papers [5,17,18,19,20,21,22,23] have alluded to as a hub for the global burden of HCV due to the number of LMICs in the region and the rising prevalence of injection drug use. For example, in Egypt, Morocco, and the United Arab Emirates, viremic rates in HCV-positive individuals in the general population were approximately 70%; the viremic prevalence varied from 0.7% (2011) in Saudi Arabia to 5.8% (2007–2008) in Pakistan and 10.0% (2008) in Egypt [17]. However, epidemiologic data from this region is lacking, and there is no consensus regarding intervention strategies that would be beneficial to eradicate HCV in this area. In this review, we briefly discuss what is known regarding HCV epidemiology in the MENA region, then focus on strategies that can be implemented to eliminate HCV in the area.

## 2. HCV Epidemiology in the MENA Region

The prevalence of viral hepatitis is generally higher in Middle Eastern countries than in Europe and the USA, possibly due to the migration of infected Asian persons [24]. Of the 560 million people in 20 countries in the MENA region (8% of the world’s population), the overall pooled mean viremic rate is 67.6% (95% CI: 64.9–70.3%), but the prevalence of HCV infection in each country varies. The prevalence was very low (<1%) in Iran, Cyprus, Saudi Arabia, Sudan, and Lebanon [25,26]; low (<1.5%) in Djibouti, Kuwait, Oman, Qatar, and the UAE; moderate (1.5–3.5%) in Algeria, Iraq, Lebanon, Libya, Morocco, Saudi Arabia, Sudan, Syria, and Tunisia; and high (>3.5%) in Egypt, Pakistan, and Yemen. In Pakistan, the HCV prevalence decreased from 6.7% in 2007–2008 [17] to 4.2% (7,001,000 cases) [27] in 2015. The highest prevalence of HCV infection, not just in MENA, but in the world, is in Egypt. The prevalence was 17.5% [28] in 2013, but decreased to 10% in 2015 [29] due to increased screening initiatives and governmental treatment programs.

There is tremendous genotype diversity in the MENA region. Chronic HCV infections are highest among genotype 3, followed by genotypes 4, 1, and 2 [20]. Genotype 1 is dominant (≥50% of HCV infections) in Algeria, Iran, Morocco, Oman, Tunisia, and the UAE, but is distributed ubiquitously across the MENA region. Genotype 2 is common (10–50%) in Northern Africa (Algeria, Libya, and Morocco) and Bahrain. Genotype 3 is the most common in Afghanistan and Pakistan, and genotype 4 is predominant in Egypt, Iraq, Jordan, Palestine, Qatar, Saudi Arabia, and Syria. Genotypes 5, 6, and 7 are essentially absent.

## 3. Risk Factors for HCV Transmission

Before screening for HCV became available, HCV infection was transmitted mainly by transfusion of contaminated blood or blood products. The other modes of exposure are healthcare-related exposures (59.5%), community-related exposures (15.8%), sexual-related exposures (3.7%), surgical and other medical procedures, dental work, and medical injections [20,30]. In the MENA region, the major risk factors for HCV transmission have been blood transfusions and hemodialysis due to the prevalence of beta-thalassemia major and end-stage renal disease. In addition, healthcare also appears to be the driver of prevalent (and possibly incident) infections in MENA, followed by injection drug use [20].

However, the global trend of transmission has shifted such that contaminated needles, syringes, and other instruments for injections and skin-piercing procedures harbor enormous risk. Among high-risk populations, PWIDs carry the highest prevalence of HCV seropositivity [31]. Transmission of HCV among PWIDs is greatest with “direct sharing” of needles and syringes but may also occur indirectly via sharing of injection paraphernalia, such as syringes, cookers, and cotton filters [32,33,34]. Approximately 20% to 30% of PWIDs become infected with HCV within the first 2 years of injecting drugs and 50% within 5 years [35].

Regarding health care access, even though persons with substance use disorders have the highest HCV prevalence and incidence, the vast majority have not engaged in care for the infection. Previously, interferon-based treatments, with substantial side effects and the propensity to exacerbate mental health conditions, were the major disincentives to pursue care for the infection. Data show that only 2% to 20% of PWIDs with HCV infection are linked to care [36]. Direct-acting antivirals, with viral eradication rates of >90%, significantly improved side effect profiles and shortened treatment duration; these dramatic advances over prior treatment regimens should promote widespread hepatitis C virus care among persons with substance use disorders [36]. Research shows that HCV treatment has been delivered successfully to PWID through various multidisciplinary models such as community-based clinics, substance abuse treatment clinics, and specialized hospital-based clinics. Models may be integrated into primary care—all under one roof in either addiction care units or general practitioner-based models—or can occur in secondary or tertiary care settings. Additional innovative models include directly observed therapy, peer-based models, and maintenance on opioid substitution therapy (OST) with methadone or buprenorphine [37,38,39]. It is important to note that OST is associated with sustained virologic response (SVR) rates of 94% to 97%, regardless of ongoing drug use [40,41]. However, PWIDs successfully treated with DAAs must be followed and tested frequently for reinfection. A recent study of collocation of HCV care in harm reduction centers allowed active PWID to engage in HCV treatment, OST and needle syringe programs, and pre-exposure prophylaxis (PrEP) for HIV. SVR rates were high in this population, and high-risk behaviors for HCV reinfection were reduced [42].

## 4. Injection Drug Use and Substance Use Disorders in the MENA Region

Due to the stigma associated with substance use, PWIDs in the MENA region are difficult to access for epidemiologic and treatment purposes [16]. The prevalence of injection drug use (IDU) in the MENA region is rising [16,43], estimated to be 0.24 per 100 adults (626,000 persons) [44]. IDU seems to be mostly among men, but women who inject drugs are difficult to reach, so epidemiologic data may be skewed. However, injection drug use among women may be related to sex work or having a sexual partner who is a PWID [44]. Though data are limited, the prevalence of HCV in PWIDs is increasing. In 2006, although the overall weighted prevalence of HCV in Iran was 0.5%, the rate was significantly higher in men (1.0%) than in women (0.1%). In multivariate analysis, male sex, history of intravenous drug abuse, and imprisonment were significantly associated with HCV [45]. In areas such as Lebanon and Syria, the HCV prevalence ranged from 40–61% among PWIDs [44]. A pooled analysis suggests that half of PWIDs in the MENA region have been infected with HCV [46]. Substance use disorders from alcohol and marijuana are prevalent in the Middle East, but in the MENA region, heroin is the most reported injectable drug [16]. Substance use has been a neglected area of public health, though it has been recognized as an important risk for SUD-related harms, including HIV- and HCV-associated morbidity [16]. HCV infection exacerbates the morbidity and mortality associated with these agents, as the hepatotoxic effects of these agents can rapidly progress to liver fibrosis. Since highly marginalized individuals such as people who inject drugs are the predominantly affected populations, innovative strategies are therefore required to enhance screening and linkage to care for so-called “hard-to-reach” populations. Several reviews and models have demonstrated that targeting PWIDs for HCV treatment is imperative to eradicating HCV globally [47,48,49,50,51]. Thus, it is of paramount importance that appropriate collocated strategies are developed and employed in the MENA region, to clinically manage PWIDs and HCV infection.

## 5. Intervention Strategies for the MENA Region

Hepatitis C elimination campaigns in lower–middle-income countries have been successful by targeting several key intervention points along the HCV care continuum. However, the substantial out-of-pocket costs for the treatment present an important barrier to service access. For example, patient assistance programs, combined with generic brands of newer hepatitis C treatment, offer a significant cost reduction and widen access to hepatitis C treatment in low- and middle-income countries [52,53,54]. Egypt was particularly successful with a national hepatitis C strategy [55] that provided no-cost DAA-based HCV treatment to more than 2 million people between 2014 and 2017. In 2018, Egypt introduced a national screening initiative that screened an impressive 49.6 million people in 7 months (79.4% of the target population). Using Egypt’s initiative as an exemplary model, we propose several interventions that must be implemented in the entire MENA region and outline those below.

Nationwide Mandate for Testing: The strategy of seek, test, treat, and retain (STTR) was found to be effective in expanding the HIV care continuum [56,57]. In the United States, the Centers for Disease Control recommends that persons born between 1945 and 1965 in the U.S. are screened for HCV infection [58], based on the period when blood transfusions were widely used but not screened for infectious diseases. The MENA region should develop its recommendations based on the risk factors for HCV transmission.

Nationwide Availability for Point-of-care Testing: In the MENA region and worldwide, many remain unsure of their HCV infection status because current testing requires at least a two-step process: (1) determination of HCV serostatus, followed by (2) HCV RNA or core antigen testing to confirm active infection. Implementing point-of-care testing using HCV core antigen would reduce loss-to-follow-up and increase the engagement of PWIDs into HCV care [59,60,61,62]. Efficient testing can also be employed using novel methods/technologies, such as the mobile health vans program (A-CHESS), which allows for the implementation of several services and the collection of longitudinal data related to drug use and HCV care among people with opioid use disorders [63], and telehealth, which is cost-effective, saves time, and facilitates outreach/access to HCV treatment [64].

Public Awareness and Treatment Availability: Mobilize public awareness campaigns to destigmatize HCV screening and treatment. Surveys of hepatitis programs in LMICs have shown that one of the major challenges to scaling up hepatitis care is limited community awareness of hepatitis [65,66]. Worldwide, only 41% of LMICs have a funded public awareness campaign for hepatitis C, and less than half have spent funds raising awareness [67]. To increase public knowledge in LMICs, lessons on the importance of awareness campaigns can be taken from HIV and applied to HCV, since the risk factors for each are similar. For example, in the 1990s, people with HIV in Thailand launched a campaign for universal access to healthcare with one slogan: “Why are medications so expensive?” Within 10 years, nearly half of those who needed treatment received it under a national healthcare program, and Thailand became a widely praised success story in the global response to AIDS [68]. The impact of efficient, creative public awareness campaigns in MENA cannot be underestimated, and they are vital to eliminating the stigma of HCV in the population, especially among PWIDs.

Treatment Availability: Publicize the low-cost of DAA treatments and appeal for government-mandated subsidization [69]. The largest barrier to accessing DAAs in LMICs is the exorbitant cost. The cost of sofosbuvir is substantially lower in the Middle Eastern countries such as Egypt (EGP 900 for 12 weeks of therapy) compared to costs of USD 84,000, GBP 54,000, and EUR 25,000 in the U.S., U.K., and Spain, respectively [70]. Sofosbuvir-based regimens effectively treat all HCV genotypes, including genotype 4, which is the most common among PWIDs in the MENA region [19]. Therefore, successful HCV eradication can be achieved in the MENA region with subsidized costs of sofosbuvir-based regimens. There are little data available on the cost-effectiveness of expanded versus restrictive treatment in LMICs and PWIDs in the MENA region. However, available studies suggest that there are substantial health benefits when HCV treatment is broadly delivered, and very low drug prices are necessary for cost-effectiveness [71]. In addition, treatment of individuals who are likely to transmit the virus (PWIDs) may have a considerable impact on reducing disease incidence and prevalence, based on data from other regions [71,72].

Integration/Collocation of HCV Treatment with OST and Harm-reduction: Engagement of PWIDs in HCV care has been difficult due to stigmatization of the HCV diagnosis in the MENA region. This causes discomfort during conventional health care settings and interactions with providers, and a public lack of HCV awareness. Other factors contribute to difficulty in engaging PWIDs, including multiple comorbidities (HIV, hepatitis B virus (HBV) infection, and polysubstance addiction), physician reluctance to treat PWIDs, required sobriety for treatment, lack of engagement in OST before HCV treatment, and limited access to DAAs due to the lack of a coordinated government effort [73]. In addition to promoting engagement in care, treatment can be collocated with OST and syringe exchange programs for harm reduction, and minimization of risk for HCV re-infection and HIV infection (via PrEP). Collocated treatment centers should be modeled like the National Rehabilitation Center (NRC) in Abu Dhabi, geographically and strategically located in the MENA region where patients with SUD, mental illness, or high-risk behaviors can be effectively treated with medications to treat SUD and/or HCV while receiving robust psychosocial and peer support. These centers could prove vital to the elimination of HCV infection in the MENA region. Modeling shows that harm-reduction strategies are necessary for HCV elimination, as they would provide sustainability post-SVR and reduce the rate of re-infection [74]. Currently, many MENA countries have no explicit plans for harm reduction strategies in their national HCV elimination policies [16].

Epidemiologic Research: Increase epidemiologic research on risk factors for HCV transmission. Research is needed to further characterize the status of HCV epidemiology in the MENA region. To date, there has been little to no investigation into populations besides PWIDs who are at high risk for HCV infection, including men who have sex with men (MSMs) and sex workers. By understanding the epidemiology of HCV transmission in this region, further public health campaigns can be developed to remove the stigma associated with testing in these populations. Epidemiologic surveys would be an effective modality for determining the risk factors, incidence, and prevalence of HCV infection in these high-risk populations.

Regulatory Policies: Lobby for government-mandated safety policies for blood banks. In some parts of the MENA region, a major risk factor for HCV transmission is still blood transfusions. Therefore, it is pertinent to implement and enforce strong regulatory policies for all blood banks and hospitals, while holding them accountable for the safety of health care workers.

Multi-disciplinary Clinical Care: Employ physicians with expertise in addiction medicine, infectious diseases, gastroenterology, hepatology, other clinical specialties, nurses, social workers, and others to provide integrated clinical care as recently shown by the ASCEND [75] and ANCHOR [42] studies. In addition, task-shifting mechanisms can be employed to establish training programs for new physicians and health care providers to clinically manage these populations [52,76].

Treatment for Incarcerated PWIDs with HCV Infection: Establish treatment programs for incarcerated persons with HCV infection, and ensure they are linked to care once they enter the general population [77]. Pegylated-interferon and ribavirin regimens have been used to successfully treat HCV-infected persons in prison [78], so the use of less complicated, more effective DAA regimens may also be successful. In Australia, scale-up of DAA treatment in prisons reduced transmission, supporting a treatment-as-prevention approach in incarcerated individuals [79]. In addition, mathematical models show that treating HCV-infected prisoners with DAAs is cost-effective [80].

## 6. Conclusions

The incidence and prevalence of hepatitis C viral infection are significantly high in PWIDs in the MENA region, but only a small number receive comprehensive care aimed at HCV therapy and harm reduction. Cost-effective DAA therapies are available in the MENA region. Therefore, complete elimination of the virus can be achieved in this region if strategies target PWIDs infected with HCV. In addition, epidemiologic data on persons with other SUDs and high-risk behavior must be collected, and these populations linked to comprehensive care that includes screening for infectious diseases. Increasing screening and access to treatment will have significant economic and health care impacts in the region by reducing morbidity and mortality. Utilizing collocated care models in the MENA region will allow providers to engage and link marginalized patients, including PWIDs, to enhance comprehensive care that will improve long-term outcomes. To meet the goal of complete elimination of HCV infection in the MENA region by 2030, it is of paramount importance that the healthcare community and policymakers implement effective strategies to significantly enhance the HCV care continuum in the area.

## Data Availability

The information regarding data on each cited study can be obtained from the individual author cited.

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
