# Peer review of "Hepatitis C Virus Infection in Persons Who Inject Drugs in the Middle East and North Africa: Intervention Strategies"

_viruses, 2021, doi:10.3390/v13071363_

Round 1
Reviewer 1 Report
This manuscript was well-written and comprehensive. Only several minor points were required to improve the content of the manuscript.
- Abstract: "SUD" should be added as the abbreviation in line 12.
- I suggest the authors to use the term "people who inject drugs (PWID)" throughout the manuscript. The abbreviation of PWID was presented in Page 3, line 122, and the full-name in Page 4, line 169 can be deleted.
- Page 4, line 163: from alcohol .... marijuana are highly prevalent.
- Intervention strategies for the MENA regions: I suggest to use the subtitles for each strategy.
Author Response
We are grateful to the 1st reviewer for excellent positive comments, and accordingly have revised the manuscript with track changes. These are also mentioned in a short cover letter to Dr. Kottilil. see attached revised manuscript.

Reviewer 2 Report
The work of Jag Khalsa and Poonam Mathur review the epidemiology of HCV in the MENA region and highlight intervention strategies to attain the WHO’s goal of HCV eradication by 2030. It is of great interest as the the prevalence of viral hepatitis is generally higher in the Middle Eastern countries than in Europe and USA. It is well written with a vast bibliography It is an almost theoretical work and it would be good if some summarized diagrams could improve the visibility of the manuscript.
Author Response
We are grateful to the 2nd reviewer for excellent and positive comments and have accordingly revised the manuscript. We have added a slide summarizing the intervention strategies with subheadings. See attached.
